# Developing a web-based patient decision aid for gastrostomy in motor neuron disease: a study protocol

Rose Maunsell [1], Suzanne Bloomfield,[2] Clare Erridge,[3] Claire Foster,[1] Maggi Hardcastle,[4] Anne Hogden [5], Alison Kidd,[6] Dominika Lisiecka,[7] Christopher J McDermott [8], Karen Morrison,[9] Alejandra Recio-Saucedo,[10] Louise Rickenbach,[11] Sean White,[12] Peter Williams,[13] Sally J Wheelwright [1]

For numbered affiliations see end of article.

**Correspondence to**
Dr Sally J Wheelwright;
S.J.Wheelwright@soton.ac.uk

## ABSTRACT

**Introduction** Motor neuron disease (MND) is a progressive, incurable disease, characterised by degeneration of the nerves in the brain and spinal cord. Due to the multisystem effects of the disease, patients are faced with many complex, time-sensitive decisions, one of which is the decision on gastrostomy feeding. There are currently no published decision aids (DAs) to support patients making this decision in the UK. This study will develop and pilot a patient DA to provide evidence-based information on gastrostomy placement and feeding that is relevant to people with MND; communicate the risks and benefits associated with each option; check understanding and clarify personal values and preferences, enabling patients to make a decision congruent with their values and appropriate for them.

**Methods and analysis** A two-phase process, observing the International Patient Decision Aid Standards, will be used to develop the DA, over 24 months starting January 2019. Phase 1 will use literature reviews and stakeholder interviews and surveys to identify essential content for the DA, and explore the best way to present this. In the second phase, a prototype DA will be developed and revised using stakeholder feedback in an iterative process. Stakeholders will include individuals with MND, their carers and the healthcare professionals working with them.

**Ethics and dissemination** Ethical approval for the study has been granted by West of Scotland Research Ethics Service, reference 19/WS/0078. Study findings will be disseminated through academic and non-academic publications, conference presentations, stakeholder websites and social media. A feasibility study will follow to explore the acceptability and practicality of the DA for patients, carers and HCPs in practice and to assess whether the DA shows promise of being beneficial for the intended population.

## INTRODUCTION

Motor neuron disease (MND) is a progressive, incurable disease, characterised by degeneration of the nerves in the brain and spinal cord, leading to muscle weakness. It is estimated that up to 5000 people are living with MND at any one time in the UK, with reported incidence rates of 1.06–2.4 per 100 000 people per

### Strengths and limitations of this study

► Review of the literature will ensure that all relevant evidence is captured.
► A user-centric approach to development will result in a decision aid (DA) that addresses the needs of patients with motor neuron disease (MND).
► Using an iterative process to determine essential content for the DA and to develop the prototype may be time-consuming.
► Allowing patients the option to have a family member, friend or carer present during the interviews has the potential to influence the honesty and openness of their responses.

year.[1] Typical life expectancy from symptom onset is 3 years.[2] However, due to the heterogeneity of MND, prognosis is unpredictable, with some individuals living for many years after diagnosis and others far less.[3] Patients will experience volatile multisystem physical deterioration over the course of their disease, which may be accompanied by cognitive and behavioural change,[4] requiring them to navigate a challenging care pathway with numerous complex decisions.

Dysphagia (difficulty swallowing) is common in people with MND. Those with bulbar subtypes of the disease will typically experience dysphagia early on in their disease trajectory, with more than two-thirds of all people with MND affected in late stages of the disease.[5] Dysphagia may cause distressing coughing and choking, laboured mealtimes and aspiration, which in turn can lead to recurrent chest infections, nutritional decline and weight loss.[6]

When an individual with MND presents with the signs and symptoms of dysphagia, gastrostomy tube feeding is often recommended, based on the assumption that there will be a beneficial impact on nutritional outcome,

survival and quality of life (QOL).[7 8] Gastrostomy refers to the opening made in the abdomen, through which the feeding tube enters the stomach, and as with any surgical procedure, carries a level of risk. However, there are no randomised controlled trials that explore the efficacy of tube feeding in MND and results from other research are equivocal, weak or lacking with respect to survival, nutritional outcome and QOL, respectively.[9]

Gastrostomy feeding can have a significant impact on an individual's everyday life, with both positive and negative consequences.[10] Negative impacts include difficulties with feed administration, handling the equipment and managing supplies and storage, along with potential clinical complications and the psychological impact, such as anxiety around the tube and social isolation through loss of shared mealtimes. Potential benefits may include relief of anxiety associated with prolonged, effortful mealtimes by removing the need to eat and drink, weight stabilisation and perceived survival benefit. From the patient and carer perspective, this means the decision regarding whether to have a gastrostomy is not straightforward. To add to the complexity, there is also a time sensitivity to this decision, which was explored in ProGas,[11] a large longitudinal prospective cohort study that investigated optimum timing and method of gastrostomy for people with MND. The study found that while earlier gastrostomy may improve survival time and nutritional outcomes, some patients might delay placement or not want a tube placing at all. This is reflected in current National Institute for Health and Care Excellence guidelines for MND,[12] which state that the role of gastrostomy feeding should be discussed at an early stage but that it must be recognised that not all individuals will wish to proceed.

While there are some excellent educational resources about gastrostomy placement and feeding, including 'myTube', informed by the findings of ProGas[11] and online information provided by the MND Association,[13] there are no published patient decision aids (DAs) for this group in the UK. DAs are designed to do more than to provide information; they are usually developed for complex, preference-based decisions that require greater consideration.[14] These decisions will often have multiple options with features or outcomes that people value differently, in this case undergoing gastrostomy placement, deciding against gastrostomy or deciding to revisit the issue at a later date. Often the evidence base for each option is limited and so the best choice depends on the personal importance the individual places on the benefits, harms, burden and scientific uncertainties.[14] By presenting evidence-based information, communicating the risks and benefits associated with each option, checking understanding and clarifying personal values and preferences, a DA may help individuals with MND make a decision about whether a gastrostomy is congruent with their values and appropriate for them.[15] They can also provide an outline for conversations with family members and healthcare professionals (HCPs) and encourage a collaborative approach to decision-making.[16] It is crucial

to acknowledge the importance of involving close family members and carers in the gastrostomy decision-making process. Research has shown that carers may directly or indirectly influence patient decision-making and that patients often want to share the burden of decision-making with family and carers,[17] in some cases relying on the carer or family member in order to participate in the decision-making process.[18] For individuals with MND, communication difficulties secondary to their disease may augment their reliance on carers or family members. While the values and preferences of the patient must take priority, developing a DA that includes content for carers ensures that these individuals are well informed and able to participate fully in discussions, thus offering the patient better support in the decision-making process. It also provides support for the carer themselves, allowing them to prepare for the potential impact of the patient's decision on their own lives and providing signposting to further support and information.

The development of a DA to support shared decision-making aligns with current NHS priorities of patient-centred, value-based healthcare. Shared decision-making is a key component of NHS England's personalised care approach, laid out in their Universal personalised care model,[19 20] as part of the Long Term Plan.[21] The model is underpinned by the evidence that shared decision-making leads to more realistic expectations, a better match between individuals' values and treatment choices and fewer unnecessary interventions.[20] NHS Rightcare's value-based initiative[22] also highlights the importance of listening to patients to identify what is of value to them to ensure that their care 'delivers' on an individual level, and using this to design and deliver services accordingly. The DA will support these NHS approaches by empowering the patient to participate in shared decision-making regarding gastrostomy placement.

The study will answer the following research questions:
1. What information should be included in a DA for people with MND considering whether to have a gastrostomy, their carers and HCPs, and how is this best presented?
2. Do people with MND, carers and HCPs find a web-based DA easy to use and useful?

## METHODS AND ANALYSIS

This research protocol details the process for the development and piloting of a DA for individuals with MND, deciding whether to have a gastrostomy tube for feeding. The study will be conducted over 24 months and started in January 2019.

The study will have two phases:

Phase 1: will address the first research question by investigating what information should be included in the DA and how this is best presented.

Phase 2: the prototype DA will be developed, and the second research question will be addressed by exploring whether the prototype DA is easy to use and useful to the

**Table 1**  Patient and carer sampling frame

|  | <6 months since diagnosis | >6 months since diagnosis |
|---|---|---|
| Gastrostomy | Aim to recruit where possible | Interview: patients n=10, carers n=5<br>Beta-testing: patients n=3–4, carers n=3–4 |
| No gastrostomy | Interview: patients n=10, carers n=5<br>Beta-testing: patients n=3–4, carers n=3–4 | Interview: patients n=10, carers n=5<br>Beta-testing: patients n=3–4, carers n=3–4 |

The amber cell is highlighted and discussed in the text below to recognise that the number of participants recruited in this group is likely to be small.

end user(s). Usefulness is defined in terms of whether it meets users' needs, attitude towards using it and perceived benefit.

This will be followed by a multicentre feasibility study (phase 3) to explore the acceptability and practicality of the DA for patients, carers and HCPs in practice, and to assess whether the DA shows promise of being beneficial for the intended population. Details of which will be presented in a later publication.

## Participants

Participants will include individuals from the three key stakeholder groups involved in the decision-making process: individuals living with MND (patients), their carers and the HCPs working with them.

Purposive sampling[23] will be employed to optimise stakeholder representation, ensuring the DA content meets the needs of the UK MND population. The sampling frame will be based on time since diagnosis and gastrostomy placement (table 1). Those participants in the 'no gastrostomy' group will include patients who have declined gastrostomy. A similar number of participants will be sought in each of the green cells. Although recruitment of patients and carers in the amber cell will be attempted, it is anticipated that the number of patients and carers who fall into this group will be small.

We aim to recruit patients who are broadly representative of the MND population in terms of age, gender and subtype of MND and we will include both people who currently live alone and those who live with others. HCPs will be drawn from a variety of professions including doctors, nurses and allied health professionals.

Inclusion and exclusion criteria will be applied to potential participants (table 2). The clinical care team will be consulted to confirm whether the individual has capacity to provide informed consent. Individuals assessed as lacking capacity and their carers will be excluded because their decision-making informational needs are likely to differ significantly from those with capacity. Future research should investigate the needs of individuals with MND lacking capacity, and their care network, in gastrostomy decision making.

## Design

The methods for the two phases of the study are based on a validated model for web-based DAs[24] and the original 2005 International Patient Decision Aid Standards (IPDAS),[25] and will observe the Medical Research Council's guidance for the development of complex interventions.[26] Phase 1 of the study will elicit the information that should be included in the DA and identify the best way to present it. In phase 2, the DA will be developed

**Table 2**  Inclusion and exclusion criteria

| Inclusion criteria | Exclusion criteria |
|---|---|
| Individuals with a confirmed diagnosis of MND with a life expectancy of at least 1 year as reported by the clinical team<br>OR<br>Carers, defined as 'Any adult who looks after a family member, partner or friend with MND. The care they give is unpaid, and they have face-to-face contact with the person with MND at least 3 times a week' (currently or within the last year). They may or may not be in receipt of carers allowance<br>OR<br>Healthcare professionals working with individuals with MND, involved in the gastrostomy decision-making process or supporting patients with gastrostomy feeding<br>AND<br>Able to communicate, either verbally or using other means, for example, using a communication aid, at phrase or sentence level | Unable to participate in English<br><18 years<br>Individuals assessed as lacking capacity by the clinical team<br>Carers of individuals assessed as lacking capacity |

MND, motor neuron disease.

and revised. A study advisory committee (SAC) including individuals living with MND, their carers and HCPs and researchers with experience in MND and/or patient decision-making has been assembled and has contributed to the study design.

## Phase 1: Content and presentation
### Literature reviews
A literature review to identify qualitative studies that explore the informational needs of people with neuro-degenerative disease considering a gastrostomy will be conducted using standard methods.[27] The search will be limited to decision-making around gastrostomy, rather than including all healthcare decisions for this group, primarily because of the invasive nature of the proce-dure. The search will also be limited to studies of patients with neurodegenerative disease, rather than any condi-tion where tube feeding may be offered, because of the parallels with MND experience and disease outcomes. Thematic analysis will be used to identify information that is valued when decision-making.

A second literature review will identify theories that could be used to underpin the DA and also the best methods for presenting information within the context of a web-based DA. This review will search for web-based DAs and studies describing their development and eval-uation, both for patients with MND and other patient groups. Methods for presenting information will be tabu-lated, along with any information about efficacy. Infor-mation about the theoretical underpinning of DAs will also be collated to help with the selection of a theory to underpin the DA and to identify the active ingredients of the decision-making process. Final theory selection will be driven by the findings from the interviews. For example, if people want to get the 'gist' in the first layer of information, then delve deeper for more detail at their discretion, the Fuzzy Trace Theory[28] may be applied. The intention is to publish both literature reviews in a sepa-rate publication(s).

### Interviews
Cross-sectional, semi-structured in-depth interviews with patient, carer and HCP participants will explore rele-vant personal experiences (individuals and carers only); important information to include when making a decision about whether to have a gastrostomy; what a DA might look like and when and how a DA should be introduced. The informational needs of carers supporting individuals to make this decision will also be investigated. An inter-view guide, developed with support from the SAC, will be used to guide but not to constrain the interviews. Inter-views will be carried out either in person or by telephone, at a time and place convenient to the participant. Inter-views will be audio recorded and transcribed verbatim. It is recognised that a significant proportion of patient participants will have communication difficulties.[29] To minimise the burden of the interviews for patients, they will be sent the interview guide ahead of the interview, to

allow them to begin to construct their answers. Patients may respond verbally or by using an alternative form of communication. This may include electronic communi-cation devices, such as a smartphone or tablet device,[30] or in writing (e.g. in person or via email). Patient and carer interviews will be conducted independently, although patient participants will be able to have a family member, friend or carer present in the interview if preferred.

Clinical (patients only) and demographic data will also be collected, during or prior to the interview, to allow description of the study sample and evaluation of whether it is representative of a typical MND population in the United Kingdom. Patient data will be obtained from medical notes, while demographic data from carers and HCPs will be reported by the participant. This will include:

Patients: age, gender, ethnicity, level of education, employment status, domestic status, family history of MND, date of diagnosis, type of MND, site of onset, stage of disease, gastrostomy placed (yes or no, date if appli-cable and why).

Carers: age, gender, ethnicity, level of education, employment status, family history of MND.

HCPs: years qualified, job title, time working in the field of MND.

Sample sizes for the interviews were chosen based on previous similar projects that have achieved data satura-tion with these sample sizes.[31] We will evaluate this assump-tion during the analysis phase by checking that the final three interviews did not generate any new themes, if this criterion is not met we will continue to carry out inter-views until it is met.[32]

### DA content
Interview and literature review outputs will be synthesised using a framework analysis approach.[33] One researcher (RM) will code each transcript, line by line, with a second researcher (SW) independently coding 10% of the tran-scripts. The coding system will be harmonised through discussion and the agreed coding system will then be applied to the whole data set, with refinement as needed. Themes will be identified by grouping together similar codes. The coding system, along with the themes iden-tified from the transcripts and the first literature review, will also be discussed with the SAC and modified if neces-sary, until consensus is reached. NVivo 12 will be used for data management.

The identified themes will be presented in a survey as pieces of information which could be included in the DA. The survey will be sent to the participants who were inter-viewed, as well as members of the SAC and study team, who all will be asked to prioritise each suggested piece of content using the prioritisation model MoSCoW.[34] For each piece of content, respondents will be asked to rate how crucial it is for this to be included in the DA (M—Must have; S—Should have; C—Could have; W—Would like if time permits). Respondents will choose whether to

complete the survey online, on a hard copy or over the telephone.

Finally, a systematic search of the evidence base relating to each suggested piece of content will be carried out to ensure all information included in the DA is accurate. The search will include published papers, guidelines, best practice documents, other grey literature and input from clinical experts on the SAC.

## Phase 2: Prototype development and piloting of the DA

Results from phase 1 will inform the development of the prototype DA. The SAC and research team will use the findings from the interviews to select the relevant theory from those identified in the literature review to underpin the DA. The SAC and research team will develop a prototype DA, using an iterative process, by incorporating each piece of information receiving the highest prioritisation scores in phase 1 survey and observing the 2005 IPDAS checklist.[25] The IPDAS checklist details required content to ensure that information about options is presented with sufficient detail for decision-making, that information is presented in an unbiased and understandable way and that methods for clarifying and expressing patients' values are included, along with tools to support making the actual decision and communicating this decision to others.

The DA will be web-based, programmed using responsive web design, enabling use on tablets, laptops and smart phones. The prototype will then undergo alpha testing, where phase 1 study participants will review the prototype and complete a survey to provide feedback on:
1. Clarity of prototype information.
2. Presentation and functionality of prototype information (user-friendliness and acceptability).
3. Perceptions of utility.

It is recognised that between the interview and alpha testing (expected time frame of 3–8 months depending on when the participant is recruited), a proportion of patient participants may become too unwell to participate or will die. The period between interview and alpha testing will be kept as short as possible; however if a patient is unable to participate in the alpha testing, this will be treated as missing data.

Feedback will be discussed with the SAC and a second version of the prototype DA will then be produced. Version 2 will undergo beta testing; newly recruited patients, carers and HCPs (n=5 of each) will talk through their use of the DA following the 'think aloud' method.[35] A think aloud interview guide will be used to prompt participants to verbalise their thoughts as they use the DA, allowing the assessment of usability, satisfaction and acceptability.[35] Immediately after this, evaluation questions will be used to ascertain users' attitudes towards using the DA, whether it meets their needs and whether they think it is of benefit.

The SAC and study team will then meet again to review the results and produce version 3 of the prototype, which will undergo beta testing with a new set of participants

(n=5 of each group). Finally, the SAC and study team will decide whether any additional changes are required and the final prototype will be produced.

The formal evaluation of the DA will take place in phase 3, the details of which will be presented in a subsequent publication. In brief, the DA will be used in clinical settings. An evaluation framework, comprising measures of both the quality of the decision-making process and the quality of the decision itself, will be developed and used to compare the outcomes of patients who use the DA and those who do not. This will be supplemented with semi-structured interviews with stakeholders to explore acceptability and practicality of the DA, and to provide insight into implementation.

## Recruitment

Patient and carer participants will be identified by a member of their clinical care team in NHS MND and palliative care clinics across the Wessex and Sheffield regions. Permission will be obtained to share the contact details of any individual interested in participating, with the research team. Support groups, such as those run by the Motor Neurone Disease Association (MNDA), will provide another avenue for the recruitment of patients and carers. A member of the research team will attend the local support group in person to introduce the study, provide a participant information sheet to those interested in participating and request permission to make follow-up contact. A third channel for patient and carer recruitment will be advertisements placed on relevant websites and social media platforms, for example, Twitter. Interested individuals will be invited to contact the researcher, who will provide a participant information sheet and request permission to make follow-up contact. The clinical care team will be consulted to confirm the eligibility of all patient participants.

Eligible HCPs will be identified either following direct approach by the researcher, via a member of the advisory committee (who will obtain consent to share their contact details with the researcher), or through advertisements via social media and professional groups. A researcher will discuss the study with the individual, provide a participant information sheet and request permission to make follow-up contact.

All participants will be asked to provide informed consent prior to participation in the study.

## Patient and public involvement

People living with MND and their carers will be involved throughout the research process. The idea of developing a DA for people with MND considering a gastrostomy was initially presented to an MND support group for feedback and to invite members to join the research team.

Following award of the grant, people living with MND and their carers have joined the SAC; through this, they will help prioritise content for the DA, evaluate each iteration of the DA and comment on all patient and carer facing study documents. They will also advise on the

practicalities of including people with MND as research participants and have input on the recruitment strategy.

The views and insights of patient and carer participants will be central to the research and will form the majority of the study sample. Participants will be asked whether they wish to be kept up to date with the project's progress and those who do will be sent a newsletter.

## Consent

The researcher will obtain written informed consent from participants before entering into the study. For patient participants, this will include seeking permission to inform their general practitioner of their participation in the study. In the case that the individual is unable to provide written consent, consent will be indicated by the participant answering 'yes' or 'no', either verbally or via a communication device, after each statement on the consent form. The form will then be signed by the parties involved in the consent conversation.[36]

Given the sensitive nature of the topics to be discussed, the researcher will make clear that participation is voluntary, and the right to ask any questions and to decline participation at any time will be highlighted during the data collection.

## Participant confidentiality

The study will adhere to the principles of Good Clinical Practice and in accordance with current Data Protection Regulations, the General Data Protection Act 2018 (GDPR) and all other regulatory requirements, as appropriate. Necessary steps will be taken to ensure that all data remain confidential at all times. Study data will be kept for 10 years from the end of the study, in accordance with University of Southampton policy. Only members of the research team will have access to the full data set.

## Dissemination

A dissemination and impact plan will be developed in the initial stages of the study. Dissemination will be led by working collaboratively with Marie Curie and the MNDA (the Funders) and Public Policy, University of Southampton, to develop and take forward the impact activities to ensure maximum publicity and benefit. Study findings will be disseminated via multiple channels, including peer-reviewed academic publications, non-academic publications, conference presentations, stakeholder websites and social media. All participants will be notified of the outcome of the study by newsletter, press release and infographic.

Once a final prototype has been developed, a multi-centre feasibility study will be carried out to explore the acceptability and practicality of the DA for patients, carers and HCPs in practice, and to assess whether the DA shows promise of being beneficial for the intended population. The protocol for the feasibility study will be presented in a subsequent publication. On completion of the feasibility study, it is anticipated that there will be a DA that can be used in clinical practice with the aim of improving

the decision-making process and quality of decision for individuals with MND deciding whether to have a gastrostomy. It is also hoped that national and regional MND guidelines will make reference to the tool. The project therefore has the potential for multifaceted impact by advancing patient care, improving service delivery and providing useful insights for the development of DAs in the future.

## Author affiliations

[1]Health Sciences, University of Southampton, Southampton, UK
[2]Community Nursing, Countess Mountbatten Hospice, Southampton, UK
[3]MND Care Centre, Southampton General Hospital, Southampton, UK
[4]Physiotherapy, Rowans Hospice, Waterlooville, UK
[5]Australian Institute of Health Service Management, University of Tasmania, Sydney, New South Wales, Australia
[6]Carer Representative, University of Southampton, Southampton, UK
[7]School of Allied Health, University of Limerick, Limerick, Ireland
[8]Sheffield Institute for Translational Neuroscience, University of Sheffield, Sheffield, UK
[9]Faculty of Medicine, University of Southampton, Southampton, UK
[10]NIHR Evaluation, Trials and Studies Coordinating Centre (NETSCC), University of Southampton, Southampton, UK
[11]Motor Neurone Disease Association, Northampton, UK
[12]Dietetic Department, Sheffield Teaching Hospitals NHS Foundation Trust, Sheffield, UK
[13]Patient Representative, University of Southampton, Southampton, UK

**Acknowledgements** CJM is supported by the Sheffield NIHR BRC.

**Contributors** All of the named authors meet the ICMJE criteria for authorship and have contributed to the paper as follows: conception of the work: SJW; design of the work, including the study protocol, critical revision of the protocol manuscript, final approval of the version to be published: SJW, RM, AH, SB, CF, AR-S, CJM, CE, LR, DL, SW, AK, PW, MH, KM; drafting of the protocol manuscript: RM, SJW.

**Funding** This work was supported by Marie Curie and the Motor Neurone Disease Association grant numbers: MCRGS-20171219-8005 and 963-794, respectively.

**Competing interests** None declared.

**Patient consent for publication** Not required.

**Ethics approval** Ethical approval for the study has been granted by West of Scotland Research Ethics Service, reference 19/WS/0078. Participant information sheets, consent forms, recruitment advertisements and general practitioner letters were also approved. All amendments to study documents will be submitted in IRAS for HRA and REC approval.

**Provenance and peer review** Not commissioned; externally peer reviewed.

**ORCID iDs**
Rose Maunsell http://orcid.org/0000-0003-4106-6519
Anne Hogden http://orcid.org/0000-0002-4317-7960
Christopher J McDermott http://orcid.org/0000-0002-1269-9053
Sally J Wheelwright https://orcid.org/0000-0003-0657-2483

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
