## [Reviewer comments · BMJ Open]

ARTICLE DETAILS

TITLE (PROVISIONAL)	DEVELOPING A WEB-BASED PATIENT DECISION AID FOR GASTROSTOMY IN MOTOR NEURONE DISEASE: A STUDY PROTOCOL
AUTHORS	Maunsell, Rose; Bloomfield, Suzanne; Erridge, Clare; Foster, Claire; Hardcastle, Maggi; Hogden, Anne; Kidd, Alison; Lisiecka, Dominika; McDermott, Christopher; Morrison, Karen; Recio-Saucedo, Alejandra; Rickenbach, Louise; White, Sean; Williams, Peter; Wheelwright, Sally J.

VERSION 1 - REVIEW

REVIEWER	Patrick J. Dunn, PhD American Heart Association, USA
REVIEW RETURNED	21-Aug-2019

GENERAL COMMENTS	Dear Authors, Thank you for conducting research in this very important area. While I applaud your initial efforts, I do have some concerns about the rigor of your qualitative design. Please be more clear about your research questions in each phase. Also, since this is a qualitative design I would go beyond the literature review and describe the theoretical/conceptual framework you are using to develop the decision aid. It is unclear to me how the information obtained in the interviews will be transformed into insights that will inform the development and use of the decision aid. My suggestion, since this is a methods paper, is to describe in greater detail the process that you will use. Do you have a coding process? What is your methodology for creating codes, categories, themes. Also, in Phase 1 you mention that the participants will have the option of an online survey or a phone interview. It is unclear whether this is separate from the initial interview. How would the survey be used to gain the same type of insights obtained in a semi-structured interview. My suggestion is to create a table with the questions that will be asked, in the semi-structured interview, online survey, and phone interview. Again, in Phase 2 it is unclear how the information obtained will be translated into greater insights into the use and utility of the Decision Aid. My suggestion is to use a mixed-methods design and collect some data from the participants on their usage and perceptions.
---

REVIEWER	Danielle Marie Muscat University of Sydney, Australia
REVIEW RETURNED	02-Sep-2019

GENERAL COMMENTS	Introduction  - The introduction is well-written and clearly outlines the rationale for this project. Methods  - The manuscript states that ‘individuals assessed as lacking capacity and their carers will be excluded...’. Can you please clarify who will make that assessment, and how? - If the protocols for the literature reviews are not going to be published elsewhere, then further information is required here. This could be included in a figure or table, and may include (for example): eligibility criteria, information sources, search strategy, data management, selection process and data collection in accordance with the PRISMA-P. - Are the authors able to provide a copy of the interview guides referred to on Pages 9 and 10 of the manuscript? - The authors refer to a ‘communication aid’ on page 9. Can you please provide further clarification as to what this is? - It may be useful for the Authors to specify which version of the IPDAS checklist they will be using: http://ipdas.ohri.ca/using.html - The authors describe the alpha and beta testing as ‘piloting’ on page 9 of the manuscript. However, I question whether this does actually represent pilot testing. Pilot studies have been defined as ‘a version of the main study that is run in miniature to test whether the components of the main study can all work together. It is focused on the processes of the main study ... it will therefore resemble the main study in many respects’. (NIHR, 2018. Feasibility and Pilot studies https://www.nihr.ac.uk/funding-and-support/documents/funding-for-research-studies/research-programmes/PGfAR/Feasibility%20and%20Pilot%20studies.pdf). Can the authors please reflect on this, and/or modify the manuscript as necessary? Other comments  - In the introduction, the authors state that “there are no randomised controlled trials (RCTs) that explore the efficacy of tube feeding in MND and results from other research is equivocal, weak or lacking with respect to survival, nutritional outcome and QOL respectively”. As such, which data will be used in the decision aid? Will the authors communicate the quality of evidence? - In the introduction, the authors refer to decision aids being used for decisions which have ‘multiple options’. Can the authors please clarify what the options would be for this decision aid?
--

VERSION 1 – AUTHOR RESPONSE

Reviewer #1

1) Please be more clear about your research questions in each phase.

Reply

See response above.

2) Since this is a qualitative design I would go beyond the literature review and describe the theoretical/conceptual framework you are using to develop the decision aid.

Reply

Thank you for this suggestion. We have provided more detail on how the second literature review will identify theories which could be used to underpin the DA, with final selection being driven by the findings from the interviews (page8):

'A second literature review will identify theories that could be used to underpin the DA and also the best methods for presenting information within the context of a web-based DA. This review will search for web-based DAs and studies describing their development and evaluation, both for patients with MND and other patient groups. Methods for presenting information will be tabulated, along with any information about efficacy. Information about the theoretical underpinning of DAs will also be collated to help with the selection of a theory to underpin the DA and to identify the active ingredients of the decision-making process. Final theory selection will be driven by the findings from the interviews. For example, if people want to get the 'gist' in the first layer of information, then delve deeper for more detail at their discretion, the Fuzzy Trace Theory (28) may be applied. The intention is to publish both literature reviews in a separate publication(s).'

As well as adding a sentence to clarify how the literature review outputs will be used in the development of the prototype DA on page 9:

'The SAC and research team will use the findings from the interviews to select the relevant theory from those identified in the literature review to underpin the DA.'

3) It is unclear to me how the information obtained in the interviews will be transformed into insights that will inform the development and use of the decision aid. My suggestion, since this is a methods paper, is to describe in greater detail the process that you will use. Do you have a coding process? What is your methodology for creating codes, categories, themes.

Reply

Thank you for this suggestion. On page 9, we have provided the following additional information on how the framework analysis will be carried out and believe this provides clarity on how the insights from participant interviews will inform the development and use of the decision aid:

'Interview and literature review outputs will be synthesised using a framework analysis approach(31). The lead researcher will begin by identifying categories and themes in the transcripts and developing a code for each. Codes will then be discussed by the research team to ensure they are also representative of the literature review findings. Consensus will be reached through team discussion. These codes will then be applied to the whole dataset, with refinement as needed. The identified themes will be used to develop a survey to inform the selection of content for the DA. The survey will be sent to the participants who were previously interviewed, as well as members of the SAC and study team, who will all be asked to prioritise each suggested piece of content using the prioritisation model MoSCoW(32). For each piece of content, respondents will be asked to rate how crucial it is for this to be included in the DA (M– Must have; S – Should have; C – Could have; W– Would like if time permits).'

4) Also, in Phase 1 you mention that the participants will have the option of an online survey or a phone interview. It is unclear whether this is separate from the initial interview. How would the survey be used to gain the same type of insights obtained in a semi-structured interview. My suggestion is to create a table with the questions that will be asked, in the semi-structured interview, online survey, and phone interview.

Reply

Thank you for raising this point. We have re-written the sentence on page 9 explaining how the survey will be completed, removing the term 'interview' (see the paragraph above). We hope that this clarifies that there will be an initial interview, followed by two surveys; one for prioritisation and the second for feedback on the prototype decision aid.

'Respondents will choose whether to complete the survey online, on a hard copy or over the telephone.'

5) In Phase 2 it is unclear how the information obtained will be translated into greater insights into the use and utility of the Decision Aid. My suggestion is to use a mixed-methods design and collect some data from the participants on their usage and perceptions.

Reply

Thank you for this suggestion. These questions will be answered in phase 3 of the study, details of which will be provided in a later publication. We have provided more information on phase 3 in the methods section on page 3 as follows:

'This will be followed by a multi-centre feasibility study (Phase 3) to explore the acceptability and practicality of the DA for patients, carers and HCPs in practice, and to assess whether the DA shows promise of being beneficial for the intended population. Details of which will be presented in a later publication.'

Reviewer #2

1) The manuscript states that 'individuals assessed as lacking capacity and their carers will be excluded...'. Can you please clarify who will make that assessment, and how?

Reply

Thank you for raising this point. We have added a sentence into the methods section on page 7 to explain that the clinical care team will be consulted to confirm whether the patient has capacity.

'The clinical care team will be consulted to confirm whether the individual has capacity to provide informed consent.'

2) If the protocols for the literature reviews are not going to be published elsewhere, then further information is required here. This could be included in a figure or table, and may include (for example): eligibility criteria, information sources, search strategy, data management, selection process and data collection in accordance with the PRISMA-P.

Reply

Thank you for this suggestion. The plan is indeed to publish the literature reviews elsewhere – this has now been stated in the manuscript on page 8 as follows:

'The intention is to publish both literature reviews in a separate publication(s).'

3) Are the authors able to provide a copy of the interview guides referred to on Pages 9 and 10 of the manuscript?

Reply

Absolutely, the interview topic guides have been attached as appendices, which have been cited in the manuscript on page 8 and 9 as follows:

'An interview guide (see Appendix 1), developed with support from the SAC, will be used to guide but not constrain the interviews.'

'A think aloud interview guide (see Appendix 2) will be used to prompt participants to verbalise their thoughts as they use the DA, followed by a number of evaluation questions immediately after completion.'

4) The authors refer to a 'communication aid' on page 9. Can you please provide further clarification as to what this is?

Reply

Thank you for this recommendation. We have now provided more detail in this sentence for clarification. This sentence on page 8 now reads:

'Patients may respond verbally, or by using an alternative form of communication. This may include electronic communication devices, aid such as a smartphone or tablet device(30), or in writing (e.g. in person or via email).'

We have also added a reference to support the following sentence:

'It is recognised that a significant proportion of patient participants will have communication difficulties(29).'

Reference: Paynter C, Cruice M, Mathers S, et al. Communication and cognitive impairments and health care decision making in MND: A narrative review. J Eval Clin Pract 2019 Jul Epub ahead of print

5) It may be useful for the Authors to specify which version of the IPDAS checklist they will be using

Reply

We have followed your suggestion and have re-written the sentence as follows, referencing both the model and the standards we will be using:

'The methods for the two phases of the study are based on a validated model for web-based decision aids(24) and the original International Patient Decision Aid Standards 2005 (25)...

6) The authors describe the alpha and beta testing as 'piloting' on page 9 of the manuscript. However, I question whether this does actually represent pilot testing. Pilot studies have been defined as 'a version of the main study that is run in miniature to test whether the components of the main study can all work together. It is focused on the processes of the main study ... it will therefore resemble the main study in many respects'. (NIHR, 2018. Feasibility and Pilot studies <https://eur03.safelinks.protection.outlook.com/?url=https%3A%2F%2Fwww.nihr.ac.uk%2Ffunding-and-support%2Fdocuments%2Ffunding-for-research-studies%2Fresearch-programmes%2FPGfAR%2FFeasibility%2520and%2520Pilot%2520studies.pdf&data=01%7C01%7C r.evil%40soton.ac.uk%7Cb031256178e3419ea71108d72fa1e1ef%7C4a5378f929f44d3ebe89669d03ada9d8%7C0&sdata=ZWF4m5M0GTbQHFrUfAUo%2Bzp6LTOquncgg61fWLbWGnw%3D&reserved=0>). Can the authors please reflect on this, and/or modify the manuscript as necessary?

Reply

Thank you for raising this point. In this context 'piloting' refers to the DA itself, rather than pilot testing of a study to evaluate the DA. We have clarified this by amending the subheading on page 9 to 'Phase 2: Prototype development and piloting of the decision aid'.

7) In the introduction, the authors state that "there are no randomised controlled trials (RCTs) that explore the efficacy of tube feeding in MND and results from other research is equivocal, weak or lacking with respect to survival, nutritional outcome and QOL respectively". As such, which data will be used in the decision aid? Will the authors communicate the quality of evidence?

Reply

Thank you for your question. Of course, the lack of conclusive evidence in this area is one of the reasons there is such a need for a DA for this patient group. Content of the DA will be based on the evidence that is available e.g. observational studies, as is the case in the MND NICE guideline: <https://www.nice.org.uk/guidance/ng42>. In the absence of evidence, expert/consensus opinion could be utilised. User opinions and feedback will determine whether and how details around the type of studies that informs the content is provided in the DA itself. These details will be included in any supporting documentation.

8) In the introduction, the authors refer to decision aids being used for decisions which have 'multiple options'. Can the authors please clarify what the options would be for this decision aid?

Reply

Thank you for this recommendation. We have amended this sentence on page 5 and provided an explanation of the decision options for this decision aid:

'These decisions will often have multiple options with features or outcomes that people value differently, in this case: undergoing gastrostomy placement, deciding against gastrostomy, or deciding to revisit the issue at a later date.'

VERSION 2 – REVIEW

REVIEWER	Patrick J. Dunn Walden University, USA
REVIEW RETURNED	16-Oct-2019

GENERAL COMMENTS	Dear authors, Thank you for your work in this very important topic. I found your approach to be very sound and will expect the results to be meaningful. Since this is a methods paper I would suggest spending more time and in greater detail describing your methods
--

	and analytic approach. For example, you describe your purposeful sampling approach and have a very detailed interview guide in the appendix, but do not mention how the information from these interviews will be analyzed, and how that information will be used to inform Phase 2 of your design. Also, I would add more details to the rigor you will use to ensure the trustworthiness of your analysis in Phase 1. How will you know that you had reached a point of saturation in your interviews? In a similar way, I would spend more time in Phase 2 explaining how the information in Phase 1 is being used. You say the results of Phase 1 will inform the development of the prototype DA, but I would add more details here of how that will happen. Also, in Phase 2 I would add more details and rigor on how you will know that the prototype is effective.
--	---

REVIEWER	Dr Danielle Muscat University of Sydney, Australia
REVIEW RETURNED	08-Oct-2019

GENERAL COMMENTS	The manuscript is clear and well-written, and the authors have done a good job in addressing the reviewers' comments. A minor point for consideration is detailed below: In regards to the aims which have been added to the end of the introduction, consider adding further explanation in terms of the word 'useful'. e.g. 'Do people with MND, carers and health care professionals find a web-based DA easy to use and useful for making decisions about gastrostomy?'. This may also need to be addressed in the 'Methods and analysis' section.
--

VERSION 2 – AUTHOR RESPONSE

Reviewer #1

1) Since this is a methods paper I would suggest spending more time and in greater detail describing your methods and analytic approach. For example, you describe your purposeful sampling approach and have a very detailed interview guide in the appendix, but do not mention how the information from these interviews will be analyzed, and how that information will be used to inform Phase 2 of your design. Also, I would add more details to the rigor you will use to ensure the trustworthiness of your analysis in Phase 1. In a similar way, I would spend more time in Phase 2 explaining how the information in Phase 1 is being used. You say the results of Phase 1 will inform the development of the prototype DA, but I would add more details here of how that will happen.

Reply: Thank you for this suggestion. To address this, on page 9 we have added further detail across two paragraphs to explain in more detail the analysis of the interviews and how this will inform the development of the prototype decision aid to be tested in phase 2:

'Interview and literature review outputs will be synthesised using a framework analysis approach(33). One researcher (RM) will code each transcript, line by line, with a second researcher (SW) independently coding 10% of the transcripts. The coding system will be harmonised through discussion and the agreed coding system will then be applied to the whole dataset, with refinement as

needed. Themes will be identified by grouping together similar codes. The coding system, along with the themes identified from the transcripts and the first literature review will also be discussed with the SAC and modified if necessary, until consensus is reached. NVivo 12 will be used for data management.'

'The SAC and research team will develop a prototype DA, using an iterative process, by incorporating each piece of information receiving the highest prioritisation scores in the Phase 1 survey and observing the 2005 IPDAS checklist (25). The IPDAS checklist details required content to ensure information about options is presented in sufficient detail for decision-making, that information is presented in an unbiased and understandable way, and that methods for clarifying and expressing patients' values are included, along with tools to support making the actual decision and communicating this decision to others.'

2) How will you know that you had reached a point of saturation in your interviews?

Reply: Thank you for this question, to answer this we have added a paragraph to the interviews section on page 9, as follows:

'Sample sizes for the interviews were chosen based on previous similar projects that have achieved data saturation with these sample sizes(31). We will evaluate this assumption during the analysis phase by checking that the final three interviews did not generate any new themes, if this criterion is not met, we will continue to carry out interviews until it is(32).'

3) Also, in Phase 2 I would add more details and rigor on how you will know that the prototype is effective.

Reply: Thank you for this recommendation. To provide more detail on how we will evaluate the usefulness of the prototype in phase 2 we have amended the following paragraph on page 10:

'A think aloud interview guide will be used to prompt participants to verbalise their thoughts as they use the DA, allowing the assessment of usability, satisfaction and acceptability(35). Immediately after this, evaluation questions will be used to ascertain users' attitudes towards using the DA, whether it meets their needs and whether they think it is of benefit.'

We have also provided more detail on phase 3 of the study by adding the following paragraph to the end of the 'decision aid content' section, on page 10:

'The formal evaluation of the DA will take place in Phase 3, the details of which will be presented in a subsequent publication. In brief, the DA will be used in clinical settings. An evaluation framework,

comprising measures of both the quality of the decision-making process and the quality of the decision itself will be developed and used to compare the outcomes of patients who use the DA and those who do not. This will be supplemented with semi-structured interviews with stakeholders to explore acceptability and practicality of the DA, and to provide insight into implementation.'

Reviewer #2

1) In regards to the aims which have been added to the end of the introduction, consider adding further explanation in terms of the word 'useful'. e.g. 'Do people with MND, carers and health care professionals find a web-based DA easy to use and useful for making decisions about gastrostomy?'. This may also need to be addressed in the 'Methods and analysis' section.

Reply: Thank you for this suggestion. We have added a definition of what we mean by usefulness on page 6 as follows:

'Phase 2: the prototype DA will be developed, and the second research question will be addressed by exploring whether the prototype DA is easy to use and useful to the end user/s. Usefulness is defined in terms of whether it meets users' needs, attitude towards using it and perceived benefit.'

As well as explaining how we will go about evaluating this in our methods on page 9:

'A think aloud interview guide will be used to prompt participants to verbalise their thoughts as they use the DA, allowing the assessment of usability, satisfaction and acceptability(35). Immediately after this, evaluation questions will be used to ascertain users' attitudes towards using the DA, whether it meets their needs and whether they think it is of benefit.'

VERSION 3 - REVIEW

REVIEWER	Patrick J. Dunn Walden University, USA
REVIEW RETURNED	02-Nov-2019

GENERAL COMMENTS	Dear Authors, Thank you for making the requested revisions. I believe it has strengthened your paper.
--